# A Geographic Information System-Based Indicator of Waste Risk to Investigate the Health Impact of Landfills and Uncontrolled Dumping Sites

**DOI:** 10.3390/ijerph17165789

**Published:** 2020-08-10

**Authors:** Lucia Fazzo, Marco De Santis, Eleonora Beccaloni, Federica Scaini, Ivano Iavarone, Pietro Comba, Domenico Airoma

**Affiliations:** 1Department of Environment and Health, Istituto Superiore di Sanità, 00161 Rome, Italy; marco.desantis@iss.it (M.D.S.); eleonora.beccaloni@iss.it (E.B.); federica.scaini@iss.it (F.S.); ivano.iavarone@iss.it (I.I.); pietro.comba@iss.it (P.C.); 2North Naples Prosecution Office, 81031 Aversa, Italy; domenico.airoma@giustizia.it

**Keywords:** GIS-approach, uncontrolled waste, dumping site, hazardous waste, landfills, exposure

## Abstract

Uncontrolled and poor waste management practices are widespread. The global health impact of hazardous waste exposure is controversial, but the excess of some diseases appears to be consistent. The Geographic Information System (GIS, ESRI Inc., Rome, Italy) method used to estimate the waste risk exposure, in an area with many illegal waste dumps and burning sites, is described. A GIS geodatabase (ESRI ArcGIS format) of waste sites’ data was built. A municipal GIS-based indicator of waste risk (Municipal Risk Index: MRI) has been computed, based on type and quantity of waste, typology of waste disposal, known or potential environmental contamination by waste and population living near waste sites. 2767 waste sites were present in an area 426 km^2^ large. 38% of the population lived near one or more waste sites (100 m). Illegal/uncontrolled waste dumps, including waste burning areas, constituted about 90% of all sites. The 38 investigated municipalities were categorized into 4 classes of MRI. The GIS approach identified a widespread impact of waste sites and the municipalities likely to be most exposed. The highest score of the MRI included the municipalities with the most illegal hazardous waste dumps and burning sites. The GIS-geodatabase provided information to contrast and to prosecute illegal waste trafficking and mismanagements.

## 1. Introduction

Waste management is a worldwide problem. At the Sixth Ministerial Conference on Environment and Health of the 53 countries of the World Health Organization (WHO) Regional Office for Europe, waste and contaminated sites were declared among the priority areas for the environmental policy agenda to reach the goals of 2030 Sustainable Development [1]. Municipal and industrial waste disposals contributed to soil and groundwater contamination in about 38% of the contaminated sites in Europe [2]. In the US, the Environmental Protection Agency’s National Priority List (NPL) included, in January 2007, 1240 hazardous waste sites. Waste storage/treatment/disposal were present in 31.5% of NPL sites across the country, representing the main activities in contaminated sites [3]. In middle–low income countries, the burden of diseases of waste-related exposures is increasing and not sufficiently recognized [4]. In Asia, 679 areas contaminated by hazardous waste were identified in seven countries [5]. In Africa, the WHO included hazardous waste among the first three main environmental risk factors for the health of the population [6]. 

Uncontrolled and poor hazardous waste management practices are widespread in some areas of both industrialized and middle–low income countries. In addition, illegal waste practices and transboundary trade, mostly from industrialized countries to middle–low income countries, impact specific areas and populations [7].

In 2016, the WHO stated that, notwithstanding that the etiological role of hazardous waste exposure on health status of populations living near waste dumping sites is controversial, the excess of some diseases appears to be consistent and reproducible [8].

A systematic review of the epidemiological studies on populations living near uncontrolled hazardous waste sites evaluated the evidence of the association between the residences close to these sites and some specific diseases [9]. The evidence of a causal link between exposure to hazardous waste and several health outcomes was categorized as sufficient, limited and inadequate. The evidence was defined as limited for several outcomes, namely: liver, bladder, breast and testis cancers, non-Hodgkin lymphoma, asthma, congenital anomalies overall and of neural tube, urogenital and connective and musculoskeletal systems, low birth weight and pre-term birth. Evidence that exposure to oil industry waste releasing high concentrations of hydrogen sulfide causes acute adverse effects was evaluated as sufficient. For most diseases, the evidence was evaluated as inadequate [9]. The methods used to estimate the population exposed to the contaminants emitted by these sites represent a crucial issue in epidemiological investigations and were considered in the review to quantify the confidence in the results of each study [9]. An overview of the exposure assessment methods in epidemiological studies on populations living in contaminated sites highlighted that most studies included in the above-described review had used indicators of exposure, based on the residence [10]. In 28 studies, the indicator was based on the residence in municipalities or zip-code areas with or near a waste site, i.e., in US Superfund sites [11,12,13] and in Great Britain [14,15]; in other investigations, the exposed populations were defined on the basis of the residential distance from a waste site, i.e., in Kuhen et al. [16] and in the paper on waste management in Spain [17]. The main difficulties in exposure assessment are caused by heterogeneity of contaminants present in waste sites, often unknown, and the diversity of exposure pathways. The lack of this information, due to the informal and uncontrolled activities, makes the use of models particularly complex. The difficulties in detecting exposed populations, outcome selection, assessing exposure to complex and often heterogeneous cocktails of hazardous chemicals, represent the usual limitations associated with epidemiological investigations on this topic [18]. 

In Italy, several epidemiological investigations were carried out in the area of Naples and Caserta provinces, since early 2000, highlighting a sub-area most affected by the health impact of waste sites [19,20], and found a correlation between specific diseases and the waste impact indicator [21]. 

In this frame, the Italian National Institute for Health (Istituto Superiore di Sanità: ISS), under formal request of the Naples North Prosecution Office, developed a specific Geographic Information System (GIS) approach to estimate the risk of waste impact, at the municipal level, in an area impacted by illegal waste sites. The 38 municipalities are located in an area of Naples and Caserta Provinces (Campania Region in Southern Italy), interested, since the late 1980s, by illegal dumps and waste burning sites (the so-called “Land of Fires”). The role of criminal organization in setting up an illegal system of delivering hazardous waste in this area, including those produced in industries of Italy’s Northern regions, is well documented [22]. 

This paper describes the Geographic Information System (GIS) method used to estimate an indicator of waste risk exposure in the 38 municipalities of Naples and Caserta Provinces at study.

## 2. Materials and Methods

### 2.1. The Study Area

The study area is constituted by the 38 municipalities of Naples North Prosecution Office territory. The area, 426 km^2^ large, includes parts of Naples and Caserta Provinces, in Campania Region, Southern Italy. Figure 1 shows the detailed map of the municipalities and census-tracts included in the study area.

The available data on waste disposals, uncontrolled and illegal waste dumps and burning waste sites located in the study area were collected in a GIS geodatabase (ESRI ArcGIS format), in different polygonal or point features, depending on the source of data. Data enclosed by the Prosecutor during judiciary inquiries were considered. Databases of Regional Agencies and Institutes, available at December 2017–January 2018, were consulted. Data on waste disposals present in Campania Region Agency for Environmental Protection (ARPAC) and Zooprophylactic Institute of Southern Italy (Istituto Zooprofilattico Sperimentale del Mezzogiorno: IZSM) database were enclosed. In addition, waste treatment plants and disposals reported on the Campania Region website [23], were considered. Moreover, data available at the Department of Environment and Health of the National Institute for Health (ISS) database, elaborated during previous monitoring activities and Projects, particularly in the “Land of Fires”, were included. In summary, the considered data sources included the waste sites present in the territory of 38 municipalities at study, in the 2008–2017 period. Uncontrolled and illegal waste management practices have been occurring in the area since the late 1980s [22] and significant remediation actions have not been carried out in the area, at the time of the beginning of the present investigation.

Initially, 3461 sites were mapped, considering all data sources. The layers were created based on the data source and the type of waste site. All layers, built on the vector data model, have geometry and attributes. The sites present in more than one layer have been left in a single layer in which the attributes available in the other data sources have been transcribed. After dropping the repeating sites and a thorough validation, 2767 out of 3461 waste sites were selected for the present investigation (Figure 2).

Analytical data on specific environmental contaminants and chemical agents present in waste were available for about 1% of the sites. Considering the low number of sites for which this information was available, estimating the exposed population based on models of environmental contamination by chemicals emitted/released by waste sites was not possible. 

Because of this limitation, a municipal hazard risk for waste was elaborated on the basis of the population living in proximity of waste sites and the estimated hazard of the waste, following the steps described below. The ascertained presence of contaminants at high concentration (such as reported in the considered database), when available, was considered in the judgment of the potential health impact of the waste site. 

### 2.2. Hazard Risk Level Index, by Waste Site

In the first step, a Hazard Index (HI) was attributed to each of the 2767 waste sites on the basis of the information available for all sites: modality of waste disposal (i.e., illegal burning sites and dumps, controlled landfills and treatment plants, temporary storage), characteristic of the site and the type of waste. The alpha-numeric HI is composed by a number, based on potential environmental impact of the different types of waste disposals/sites (based on expert judgment), and a letter, corresponding to the type of waste. 

A panel of three experts evaluated the potential environmental impact of each category of site, on the basis of the likelihood of environmental contamination, considering the probability of leaching and/or runoff in groundwater and of emission of contaminants in air and in soil. In 1% of the sites for which the presence of contaminants (organic/inorganic) was reported, the corresponding HI value has been raised up. 

#### 2.2.1. The Numbers, Attributed on the Basis of Modality of Waste Disposal and the Characteristics of the Site, Range from 5, the Most Impacting, to 1, the Least Impacting

5: Burning waste sites.

4: No visible dumps (sunken or buried) of potentially hazardous and highly hazardous waste, pits with sunken waste, dumps with documented contamination of soil and groundwater and uncontrolled landfills. 

3: Dumps with soil contamination, spillage of sludge, hazardous waste heaps, and dumps of metal drums and pit slag heaps. 

2: Contaminated waste treatment plants, heaps of urban waste, storage and treatment plants of hazardous waste, landfills of industrial waste and contaminated disused industrial activities. 

1: Storage and treatment plants of waste, heaps of urban waste, industries with potential waste production and landfills of urban waste.

#### 2.2.2. The Letters, Attributed on the Basis of the Type of Waste (from A, the Highest Score, to F, the Lowest Score), are Described below

A: Potentially highly hazardous waste, no visible waste (sunken or buried).

B: Hazardous waste.

C: Industrial waste with possible release of hazardous substances.

D: Non-hazardous waste with possible release of hazardous substances.

E: Non-hazardous waste, in uncontrolled disposals, with possible environmental contamination.

F: Non-hazardous waste in controlled disposals and illegal heaps of undefined waste, with unlikely release of contaminants. 

F *: Non-hazardous waste in controlled storage sites and disused industries and quarries without any information. 

In addition, the letter “G” was attributed to the activities without supposed waste release (e.g., heaps of inert waste, less than 10,000 mc, no defined treatment plants, productive activities): these sites were not considered in the waste hazard risk estimate.

#### 2.2.3. The Alpha-Numeric HI

Table 1 shows the devised site-specific Hazard Index (HI) with the corresponding waste site, and the criteria for the attribution. Considering the sites located in the area at study, eleven combinations of the alpha-numeric hazard index (HI) were identified in the study area: 5A, 4A, 4B, 3B, 2B, 2C, 1C, 1D, 1E, 1F and 1F * (Table 1).

#### 2.2.4. Conversion of Alpha-Numeric Index to Numeric Hazard Risk Level

The next step was to derive a hazard quantification (Hazard Risk Level: HRL) from the alpha-numeric index. The experts’ panel attributed a numeric HRL to each HI, considering that the magnitude of the impact of the sites depends on the combination between the type of waste (the letters of the HI index) and the modalities of disposal and characteristics of the site (the number of HI). The attribution of the numeric correspondence was based on the likelihood of contamination by toxic agents and their potential hazard for the population. The most weight was attributed to the numeric part of HI, one order of magnitude moving from one to the next of the five levels, while letters were transformed in numbers without any amplification through different levels (Table 2). The conversion of the numeric part to ten powers was applied in such a way that the presence in one area of multiple low-hazard sites produced a hazard level lower than one high-hazard site (i.e., an illegal burning waste site is considered to be far more impacting than many controlled landfills of inert materials). The value of HRL is the product between the two numeric correspondences (i.e., an HI score equal to 5A corresponds to a hazard risk level of 60,000:10,000 * 6).

### 2.3. Municipal Waste Risk Index

The final aim of this study is to estimate the human health risk of populations residing in the study area due to the waste sites, at the municipal level. To reach this objective, the next step was to consider the HRL of the sites and the population living in the areas potentially impacted by the waste sites.

A new polygonal layer was therefore generated by the merge geoprocessing tool, preserving geometries and HRL of each waste site for all layers. The point layers have been previously transformed into hexes with 100 m apothem. Figure 2 shows the map of the waste sites.

In order to identify the population living in the areas impacted by the waste site, a circular buffer of 100 m radius around the features of each waste site was generated, first without the dissolve option and subsequently with this option. The intersection of these two layers generated the break buffer overlaps-up (Figure 3). 

The choice of a 100 m radius, much lower than what has been used in some comparable studies (1–2 km), is motivated by the high density of waste sites in the study area (around 3000 sites in an area of about 426 km^2^). Buffers of larger breadth have caused an overlap of the impacted areas and the whole area would appear to be affected by waste impact, preventing a distinction of the areas at different degrees of waste impact. For each municipality, we considered the waste site buffers (100 m) laying in the municipality, even if the centroid was located in other municipalities of the area at study.

The resulting layer was combined (union geoprocessing tool) with the layer of the 2383 census tracts (Figure 1) with the data relating to the resident population previously associated (Census 2011).

The union between the two layers (waste sites and census-tracts) generated about 75,000 polygons, with many overlaps despite the small size of the buffers. Territorial overlaps net, the number of polygons has fallen to around 26,000 (Figure 4).

A multi-code HRL (equal to the sum of HRL) was attributed to the areas influenced by more than one site, with an ad hoc Visual Basic software (Microsoft Office, Istituto Superiore di Sanità Licence, Roma, Italy). The population living in the areas impacted by waste was estimated on the basis of the density of population in the census-tract where the polygon falls. 

For each polygon, a Risk Index (RI) was computed:
RI = S × HRL × S/Sc × P,(1)
where S is the surface of the polygon, HRL is the hazard risk level index of the waste site, or the multi-code HRL of the waste sites, lying in the polygon, Sc is the surface of the census-tract, P is the population residing in the census-tract and S/Sc × P is the estimated population residing in the polygon. RI is proportional to the population living in the census-tract: if the buffer falls in an inhabited census-tract, the RI is equal to 0.

Subsequently, for each municipality, the areas influenced by one or more waste sites, with the corresponding HRL and living population (site-specific RI), were considered. 

A waste Risk Index at the municipal level (Municipal Risk Index: MRI) was computed, summing up the scores of all areas (polygons) comprising the municipality:
(2)$$MRI=\sum_{p=1}^{n}RI_{p}$$
where *p* is the number of polygons lying in the municipality, and *RI_p_* is the Risk Index of polygons lying in the municipality. 

Finally, municipalities were categorized into four classes of risk (1—low to 4—high), on the basis of MRI, using Jenks’ method (Natural breaks), that maximizes homogeneity within groups and variance between groups. The categorization of municipalities in 4 MRI classes was evaluated as the most appropriate, with respect to 5 or more classes, to distinguish the municipalities at the highest MRI, after a sensitivity analysis considering the distribution of municipalities and residing population by MRI classes. 

## 3. Results

The study area, constituted by 38 municipalities, is 426 km^2^ large. In this area, on the basis of the available data at January 2017, 2767 waste sites were mapped. The GIS approach allowed us to estimate, in the whole study area, 354,845 people (38% of the total population) living close to one or more waste sites (in a buffer of 100 m). In the whole area, there is a high environmental pressure by waste sites and likely human exposure to a variety of agents, including toxic ones.

Table 3 reports the distribution of waste sites by HI score and municipality.

The most represented HI score class of waste sites (33% of all waste sites) is that of HI = 1F, corresponding to the waste sites with unlikely release of hazardous substances (controlled urban waste landfills, treatment plants of non-hazardous waste and illegal heaps of undefined waste), followed by the group of waste burning sites (HI = 5A, 23%) and by dumping sites with hazardous wastes and documented or potential contamination of soil (HI = 3B, 20%). Considering the HI score classes corresponding exclusively to illegal waste sites (5A, 4A, 4B, 3B) and the uncontrolled waste disposals or heaps included in other HI score classes, on the basis of their hazard level, all illegal and uncontrolled waste sites represent about 90% of the waste sites present in the study area. All municipalities have more than one waste site. The municipalities with the highest number of waste sites are Giugliano in Campania and Caivano (628 and 282 waste sites, respectively) and Casavatore is the municipality with the least number of waste sites (3 sites). Twenty-seven percent (178 sites) of all waste burning sites (HI:5A) are located in Giugliano in Campania.

Table 4 shows the surface impacted by waste, the population living in these areas and the corresponding MRI class, by municipality. 

Considering the area of impact (100 m buffer) of waste sites falling in municipal areas, even if the centroid of the site was located outside the municipality, 7.9% (4510 inhabitants) of the municipal population and 1.3% of municipal surface represented the lowest percentage of the municipal population living in one or more waste site buffers and of the impacted area, respectively (Table 4). 

Figure 5 shows the geographical distribution of MRI classes by municipality. Class 4 includes the two municipalities with the highest waste risk for the population, while municipalities of MRI class 1 are the least impacted with respect to those at study, even if there is a waste risk exposure. 

## 4. Discussion

The adopted GIS approach [24,25] identified areas with different levels of waste risk exposure for the population, also in the lack of analytical data on environmental contamination.

### 4.1. The Main Results

The whole area at study is impacted by waste sites and 354,845 people (38% of the population residing in the study area) live within 100 m from one or more waste sites. Also, the municipalities least impacted by waste sites include a percentage of the population living near waste sites (at least around 8%), thus at risk of exposure. The municipalities with the highest MRI correspond to those with the highest number of burning waste sites and illegal hazardous waste dumps. In these areas, the implementation of remedial actions appears to be a priority. The results reported in the GIS-database permit, on the one hand, the Prosecution Office to modulate in the most updated way the investigations and their priorities, and, on the other hand, Public Health Authorities to schedule convenient precautionary strategies.

### 4.2. Selection of Waste Impact Area Buffer

A conservative method was developed, in order to highlight the most exposed populations. The radius (100 m) used to identify the population exposed to waste sites is relatively short with respect to the one of 1–2 km used in other similar investigations [9]. The rationale for choosing a 100 m radius buffer was illustrated in the Methodology Section. The choice was based on a priori considerations. One alternative approach might have been that of selecting the radius after testing the effects of the adoption of different values for this parameter. Given the context and the aims of the study, namely, to provide elements for priority evaluation in adopting site-specific remedial actions, it was agreed to follow the a priori rather than a posteriori approach. Moreover, the application of a low radius waste impact area allowed us to distinguish the subareas at different levels of waste impact and to highlight the highest impacted areas, preventing false positives. 

### 4.3. Limits of the Adopted Method

Some limitations of the adopted approach have to be discussed. The present investigation focused on the sites where mismanagement and illegal dumping and burning of waste cause a potentially dangerous exposure for the population and did not consider other potential sources of contamination. Some industrial emissions might represent a source of environmental pollution in the areas, but this issue was not an object of the request of the Prosecutor.

The municipal level of the waste indicator and the use of the distance from the waste sites as a proxy of environmental exposure, in particular, deserve some considerations. Firstly, the attribution of the same indicator of waste risk to all people living in a municipality could represent a bias in the exposure assessment process. Considering the percentage of the population living near waste sites with different levels of impact, in the computation of the indicator, though, could reduce the distorsion. The indicator of waste risk was computed at the municipal level because of the availability of health outcomes data (mortality, hospitalization) at this same geographical level, that will be used in the epidemiological study to assess the association between waste exposure and occurrence of specific diseases. The investigations carried out at the population level, applying appropriated methods, provide useful indications for public health interventions to be implemented at the community level [26]. On the other hand, the findings, informative at the population level, may not be used to infer individual exposure levels, which would lead in this case to the so-called “ecological fallacy”. In the cases of availability of outcome data at smaller areas (e.g., census-tract) or at the individual level, the described GIS method and the criteria adopted to define the potential health impact of waste sites, could be still applied. 

Second, in order to identify the exposed population, the distance of the residence from a waste site was used. In the lack of information on environmental contaminants, the distance from the source of environmental contamination may represent a reasonable indicator of environmental exposure [11,12,13,14,15,16,17]. In the contexts where large areas and populations are affected by mixtures of unknown substances potentially emitted or released, carrying out biomonitoring studies and applying dispersion models can be problematic [18]. Furthermore, because of the lack of analytical environmental data, the indicator was based on the estimated toxicity of waste and on the likelihood of the exposure to emitted/released toxic substances. Detailed environmental data, including biomonitoring data, when available, might enrich the indicator, by making it more accurate and less vulnerable to random misclassification that implies underestimation of risk. 

Considering the possible health impact of electric and electronic waste disposals [8,27,28,29], we point out that no explicit information regarding the presence of this type of waste is available in our dataset. Despite this, in view of the illegal and uncontrolled management of waste in the study area, the presence of e-waste, that was not considered, could not be ruled out. In addition, considering that the impact areas of waste sites located in neighboring municipalities, but excluded by the study area, were not considered, the exposed population at waste risk could be underestimated. 

Lastly, a limitation of the present investigation could be represented by the lack of information on the running period of each site, partially due to the informal and uncontrolled nature of the practices. Most waste sites identified in the study area exert their polluting actions starting from the early 1990s. Illegal trafficking and dumping of hazardous waste by crime organizations in a sub-area of Campania Region including that of the present study, since the end of the 1980s, are documented, on the basis of crime organizations’ exponents’ statements and judiciary investigations [22]. 

### 4.4. The Developed MRI

The meaning of the different classes of MRI deserves specific considerations. The MRI developed in this study updates the previously computed indicator [30] used in an earlier geographic mortality study [21]. The new indicator considers a smaller territory (38 versus 196 municipalities), but a higher number of waste sites (2767 versus 226), including the burning waste sites not previously investigated. The integration of several data sources on the waste sites present in the study area, including those enrolled by judiciary authorities, provided an updated and a more detailed and exhaustive database.

The Hazard Index, and therefore the classes of the related MRI, is based on expert judgment with rare availability of measurements of pollutants to validate the choice of the categories. This process might be unable to quantify differences in terms of specific hazards. However, it allows us to distinguish among different likely hazardous exposure scenarios and to highlight municipalities differently impacted by waste, when looking at the health profile of populations residing in these areas. HRL score classes represent a “relative” score, pointing out municipalities with the highest waste impact among those at study, and the municipalities included in the lowest HRL score class (class 1) are also impacted by waste sites (Table 4). The present categorization in 4 classes makes out, in particular, the municipalities at the highest waste impact (4 and 3 MRI classes), the principal aim of the present investigation. A different categorization of the absolute value of MRI is possible. 

### 4.5. The contributions of the Present Investigation

Notwithstanding the above-described limitations, the present study highlighted the need for interventions of environmental remediation in the whole area, and, primarily, in the municipalities with the highest MRIs. In particular, illegal and uncontrolled waste dumping and burning, still ongoing after more than thirty years, have to be urgently stopped.

The GIS indicator, built as described above, will be applied to search correlations with health outcomes in ad hoc epidemiological investigations. These studies will quantify the health impact of waste sites, focusing on the diseases that recognize exposure to waste among their risk factors, and will identify the diseases, also in the childhood and adolescent population, requiring the implementation of appropriate health surveillance and healthcare actions.

## 5. Conclusions

The present investigation was inspired by and originated from a formal request to the North Naples Prosecutor Office to the National Institute for Health, to identify the municipalities at the highest waste impact in an area characterized by a widespread waste mismanagement and illegal practices. The ad hoc-developed GIS-approach, notwithstanding the limitation discussed in the paper, identified the municipalities affected by different waste impact levels. 

In the whole area, there is a high environmental pressure by waste sites and likely human exposure to a variety of agents, including toxic ones. Thirty-eight percent of the population lives within 100 m from one or more waste sites. Illegal and uncontrolled waste sites, including 653 waste burning sites, represent about 90% of all 2767 waste sites present in the area. Municipalities with different levels of predicted waste exposure risk for the population were identified with the GIS approach. 

On the basis of the present results, environmental remediation actions and stopping still ongoing illegal and poor waste management practices are urgently needed. The data reported in the GIS system provide useful information to the Prosecutor, in order to contrast illegal waste mismanagements and to prosecute the criminal acts regarding waste trafficking and management. The MRI will be applied in further epidemiological study on the correlation between waste impact and the occurrence of specific diseases, in the population residing in the study area.

The GIS method described in the present paper could also be usefully applied in the lack of detailed environmental data and analytical information on waste sites. These situations could be common in similar contexts, of informal and illegal waste management, in specific areas of both industrialized and middle–low income countries.

## Figures and Tables

**Figure 1 ijerph-17-05789-f001:**
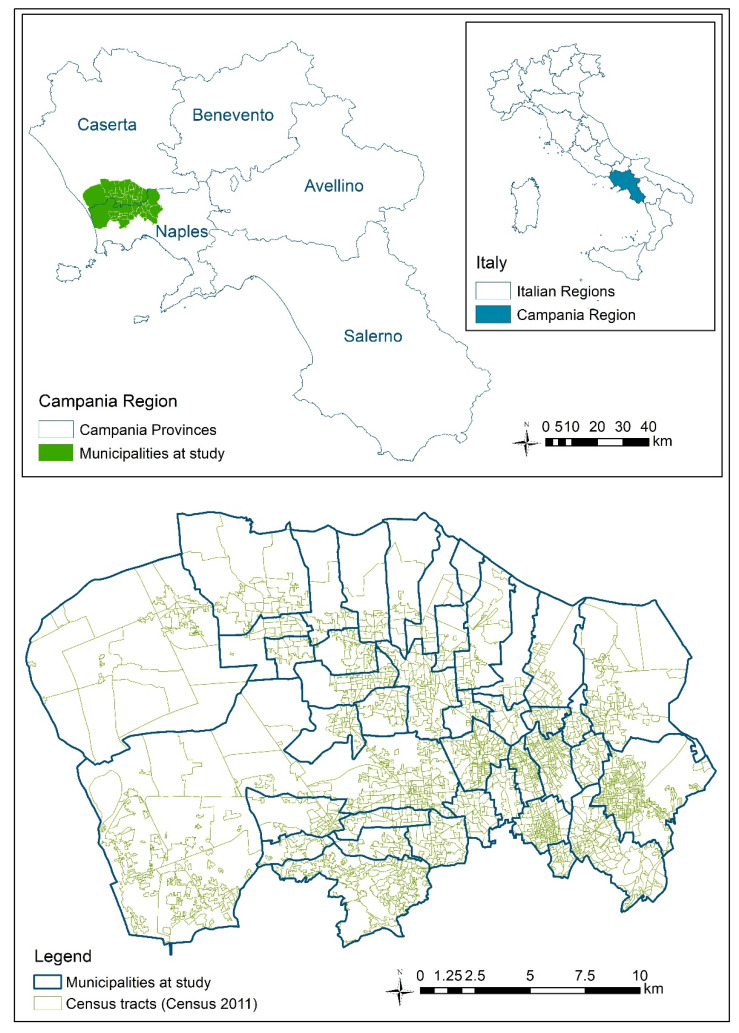
The municipalities at study (Campania Region, Sothern Italy).

**Figure 2 ijerph-17-05789-f002:**
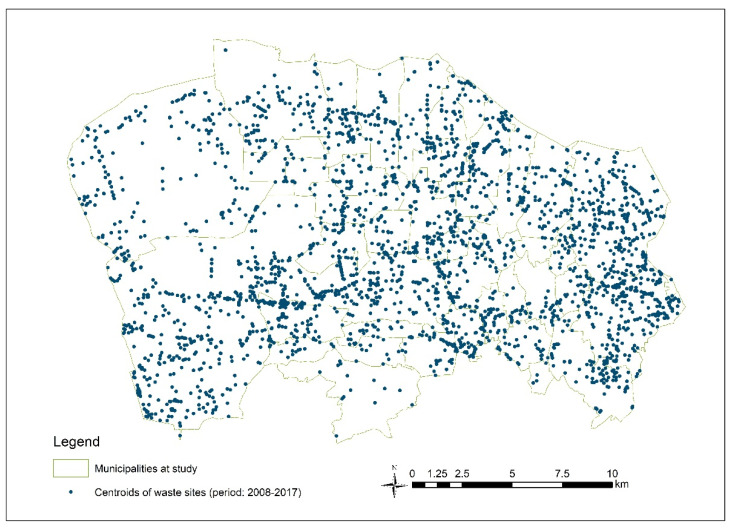
The waste sites in the study area. Period: 2008–2017.

**Figure 3 ijerph-17-05789-f003:**
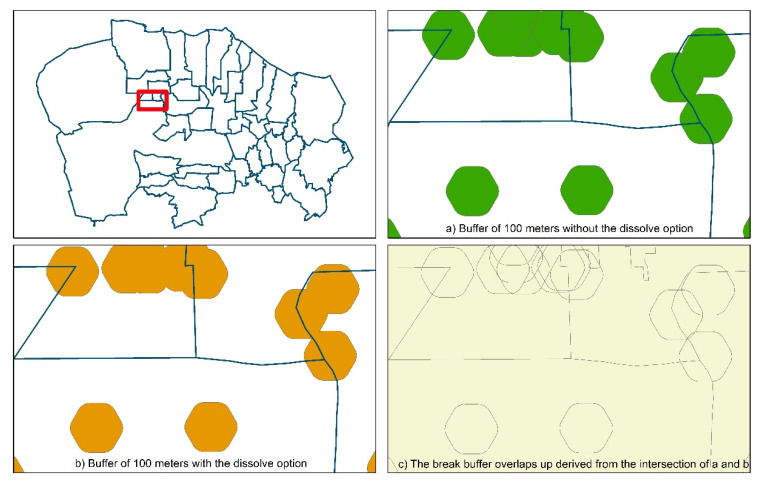
The steps for creating the break buffer overlaps up (detailed view): (**a**) buffer of 100 m without the dissolve option; (**b**) buffer of 100 m with the dissolve option; (**c**) the break buffer overlaps up derived from the intersection of (**a**,**b**).

**Figure 4 ijerph-17-05789-f004:**
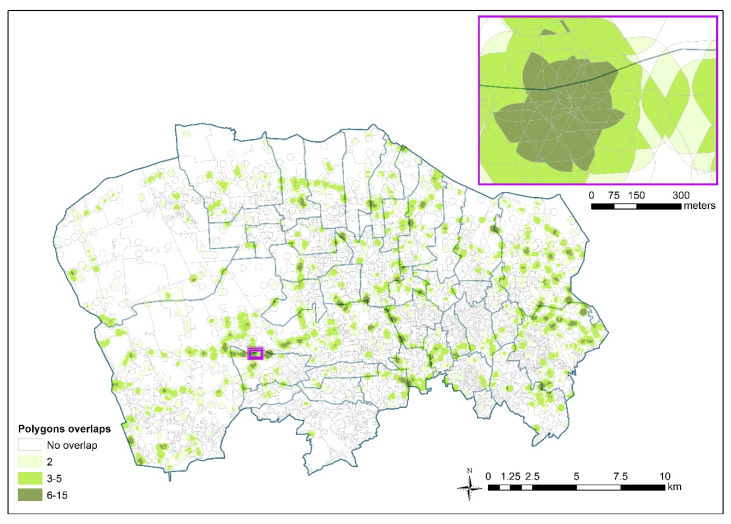
Overlaps of waste site impact areas after the union between the census-tracts layer and the break buffer overlaps-up layer. Detailed view in the purple frame.

**Figure 5 ijerph-17-05789-f005:**
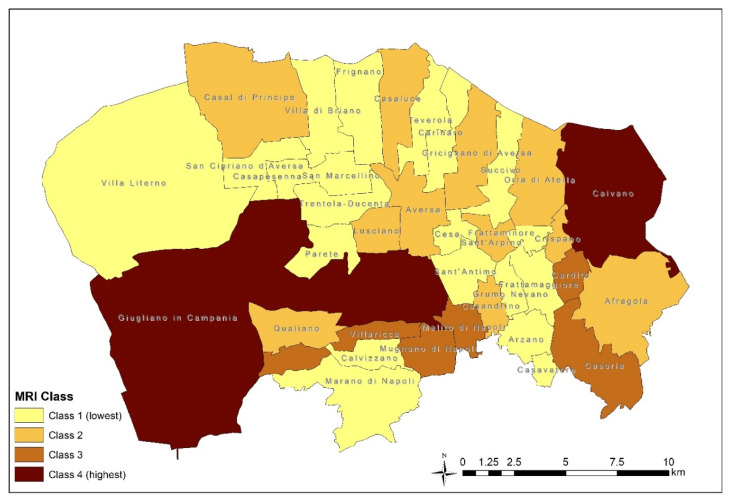
Distribution of Municipal Risk Index (MRI) classes by municipality.

**Table 1 ijerph-17-05789-t001:** Site-specific hazard index (HI), by waste site.

Hazard Index (HI)	Waste Site	Rationale
5A	Burning of waste, plastic and temporary waste storage	Possible contamination of all environmental media (air, soil, water) and presumed presence of high hazardous waste.
4A	Potentially very hazardous and hazardous non-visible waste (sunken or buried), quarries with sunken waste	Possible contamination of soil and groundwater. Considering that the wastes are non-visible, we presumed that they are hazardous or very hazardous.
4B	Illegal dumps and uncontrolled landfills with contaminated groundwater and soil	Presence of hazardous waste, with contamination of soil and groundwater.
3B	Illegal dumps with documented contamination of soil, spillage of sludge on contaminated soil, illegal heaps of hazardous waste (asbestos, tires, household appliances, paints, also volume < 10,000 mc), dumps of metallic drums, hazardous and no hazardous waste in quarry	Hazardous waste, with potential contamination of soil.
2B	Waste treatment plants with documented contamination, illegal heaps of urban and undefined waste, storage and treatment disposals of hazardous waste	Documented or potential contaminated sites with hazardous and undefined waste.
2C	Illegal heaps/dumps of special non-hazardous waste, landfills of special/industrial waste, disused industrial activities with documented environmental contamination	Sites with special/industrial waste with possible release of hazardous substances.
1C	Car wrecking and scrapping plants, electric and electronic waste reclaim plants, treatment plants of spills and special waste	Plants that, if well managed, are not supposed to generate environmental impact. Waste with prevalent inorganic compounds, and possible release of dangerous substances.
1D	Temporary storage of non-hazardous waste or organic substances, illegal heaps (>10,000 mc) of non-hazardous and urban waste, plants of recycling and reuse of specific substances, landfills of urban waste and biogas plants with documented contamination	Non-hazardous waste with predominant organic compounds in contexts with a risk of potential release of hazardous substances.
1E	Uncontrolled landfills of urban waste, dumps of non-hazardous waste, illegal heaps of urban (<10,000 mc) and inert waste (>10,000 mc)	Non-hazardous waste in uncontrolled disposals, with possible environmental contamination.
1F	Controlled landfills of urban waste, plants of waste treatment, plants for refuse-derived fuel selection and production, refluent-water depuration plants, plants of organic substances reuse, illegal heaps of undefined waste	Non-hazardous waste in controlled disposals and illegal heaps of undefined waste, with unlikely release of hazardous contaminants.
1F *	Temporary waste storage sites, disused industries and quarries, without any information	Waste storage sites and activities of undefined typologies, with possible environmental impact.

**Table 2 ijerph-17-05789-t002:** Conversion table from alpha-numeric score to the corresponding Hazard Risk Level (HRL).

Numeric Part of HI	Numeric Correspondence in Terms of Relative Potential Waste Impact	Alpha-Part of HI	Numeric Correspondence in Terms of Relative Potential Waste Impact
1	1	A	6
2	10	B	5
3	100	C	4
4	1000	D	3
5	10,000	E	2
		F	1
		F *	0.5
		G	0

**Table 3 ijerph-17-05789-t003:** Municipal distribution of waste sites, by hazard index.

Municipality	Alpha-Numeric Hazard Index Score												
	5A	4A	4B	3B	2B	2C	1C	1D	1E	1F	1F *	1G	Total
Afragola	88			47	13		3	2	10	64	1		228
Arzano	6			3		1	1	7	1	14	1	1	35
Aversa	15			10	4		15	3	13	25	2	10	97
Caivano	85			52	6	1	7	25	13	83	3	7	282
Calvizzano	2			2					1	5	1		11
Cardito	14			2			1	1	2	7			27
Carinaro	4			11	7		1	2	6	18		2	51
Casal di Principe	10			20	5				9	40		5	89
Casaluce	6			8	3		3	1	3	14		1	39
Casandrino	10			12	1		1	1	7	10	1		43
Casapesenna	1			2						4		1	8
Casavatore							1			1		1	3
Casoria	21		2	13	3	2	3	6	6	25			81
Cesa	4			12			1	1	3	10		2	33
Crispano	6			4			2	1		5			18
Frattamaggiore	7			4						15			26
Frattaminore	1			2						2			5
Frignano	5			22	2				2	9	2	4	46
Giugliano in Campania	178	7	3	113	27	2	12	13	41	193	21	18	628
Gricignano di Aversa	17			20	8			7	5	37		5	99
Grumo Nevano	3			5				2		8			18
Lusciano	8			11				1	3	11			34
Marano di Napoli				4		2		2	1	6	1		16
Melito di Napoli	20			7	1	1		1	6	9	1	12	58
Mugnano di Napoli	12						1	2	5	22			42
Orta di Atella	13			16	2			5	8	24	1	8	77
Parete	2			14		1	1	1	3	11			33
Qualiano	26			8	2	4	7	1	3	33	2	3	89
San Cipriano d’Aversa	3			4	1	1			2	7			18
San Marcellino	6			3	1			1	4	20			35
Sant’Antimo	15			18	2			3	5	17	1		61
Sant’Arpino	5			5			3	1	1	6			21
Succivo	2			13	4		1		2	12	1		35
Teverola	9			13	3			1	5	16	1	4	52
Trentola-Ducenta	5			17	2		4		4	16		2	50
Villa di Briano	5			26	7				7	13		1	59
Villa Literno	29		1	16	5		4	4	16	93		4	172
Villaricca	10		1	6		1	5	1	2	11		11	48
**Overall area**	**653**	**7**	**7**	**545**	**109**	**16**	**77**	**96**	**199**	**916**	**40**	**102**	**2767**

In bold—the waste sites in overall area at study.

**Table 4 ijerph-17-05789-t004:** Municipal Waste Risk Index (MRI) class, by municipality.

Municipality	Total Surface (km^2^)	Total Resident Population(Census 2011)	Impacted Surface (km^2^)	Population Living in Impacted Areas	% Impacted Surface	% Population Living in Impacted Areas	MRI	MRI Class
Afragola	17.9	63,820	6.9	31,446	38.7	49.3	7,216,280,909,910	2
Arzano	4.7	34,933	2.0	13,792	42.4	39.5	2,906,783,130,780	1
Aversa	8.9	52,830	3.1	22,165	35.5	42.0	9,079,920,308,650	2
Caivano	27.2	37,654	12.0	25,025	43.9	66.5	29,193,511,853,900	4
Calvizzano	4.0	12,537	1.0	2871	25.5	22.9	2,043,298,886,580	1
Cardito	3.2	22,322	1.8	11,725	55.8	52.5	10,594,357,834,900	3
Carinaro	6.3	6886	0.7	4195	10.4	60.9	487,496,802,633	1
Casal di Principe	23.5	20,828	6.3	5933	26.6	28.5	8,530,965,119,190	2
Casaluce	9.6	10,001	1.9	3557	19.5	35.6	7,054,783,196,540	2
Casandrino	3.2	13,295	1.9	6385	59.6	48.0	3,992,664,314,030	2
Casapesenna	3.0	6651	0.6	1434	20.8	21.6	1,513,540,601,150	1
Casavatore	1.5	18,663	0.3	5430	22.4	29.1	2,481,882,968,400	1
Casoria	12.1	78,647	4.7	28,294	38.5	36.0	11,242,208,588,300	3
Cesa	2.7	8496	0.7	6289	25.3	74.0	3,159,027,823,360	1
Crispano	2.2	12,411	1.3	7889	57.3	63.6	5,435,558,116,630	2
Frattamaggiore	5.4	30,241	1.5	3224	27.0	10.7	382,622,392,175	1
Frattaminore	2.0	15,708	0.2	1813	8.3	11.5	2,690,135,078,300	1
Frignano	9.9	8733	2.9	1894	29.1	21.7	52,070,765,307	1
Giugliano in Campania	94.6	108,793	37.8	49,992	40.0	46.0	47,991,495,125,500	4
Gricignano di Aversa	10.0	10,559	0.9	6059	8.6	57.4	5,860,367,569,220	2
Grumo Nevano	2.9	18,017	0.8	3496	26.1	19.4	411,167,420,981	1
Lusciano	4.6	14,539	1.9	4570	41.5	31.4	4,053,100,891,570	2
Marano di Napoli	15.7	57,204	1.5	4510	9.5	7.9	23,645,917,873	1
Melito di Napoli	3.8	36,933	2.4	22,524	62.9	61.0	20,595,282,232,400	3
Mugnano di Napoli	5.3	34,504	2.0	10,104	38.5	29.3	11,182,290,114,800	3
Orta di Atella	10.8	24,796	4.8	7852	44.4	31.7	4,615,571,979,040	2
Parete	5.6	11,012	2.0	2637	35.5	23.9	2,622,365,433,140	1
Qualiano	7.4	24,744	3.5	7226	47.6	29.2	5,006,981,801,640	2
San Cipriano d’Aversa	6.2	13,416	0.4	2563	7.1	19.1	2,861,131,558,290	1
San Marcellino	4.6	12,643	2.1	5058	45.8	40.0	3,366,623,101,160	1
Sant’Antimo	5.9	34,107	2.4	9538	41.4	28.0	2,589,050,275,760	1
Sant’Arpino	3.2	14,076	1.6	6756	50.4	48.0	5,933,015,486,750	2
Succivo	7.2	8148	0.1	784	1.3	9.6	3,610,027,624,220	1
Teverola	6.7	13,610	3.4	4416	50.1	32.4	1,769,188,408,610	1
Trentola-Ducenta	6.7	17,797	2.6	5786	38.8	32.5	440,879,467,158	1
Villa di Briano	8.5	6066	2.6	1619	30.9	26.7	1,483,028,892,830	1
Villa Literno	61.8	10,715	11.8	5445	19.0	50.8	2,804,003,658,030	1
Villaricca	6.9	30,052	2.5	10,546	36.6	35.1	10,031,129,145,600	3
**Overall area**	**426.0**	**956,387**	**136.8**	**363,427**	**32.1**	**38.0**

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
