# Peer review of "A Geographic Information System-Based Indicator of Waste Risk to Investigate the Health Impact of Landfills and Uncontrolled Dumping Sites"

_ijerph, 2020, doi:10.3390/ijerph17165789_

Round 1

Reviewer 1 Report

This paper developed a GIS-based approach for assessing the impact of waste sites on the nearby population. With this approach, the population and regions affected by waste sites were identified by using a municipal risk index. Overall, this paper is well presented. A solid background has been provided, a reasonable methodology has been used, and the results have been interpreted. However, there are still some problems regarding writing, paper structure, and methodology. The detailed comments are as follows:

1 Writing

a). Too many long and complex sentences, which affect the readability of this paper, such as lines 173-176 and lines 366-370.

b). Minor issues with grammar and the use of collocations. For example, line 47-48, ‘illegal waste practices and …, impact to specific areas and populations [7].’, either use ‘impact on’ or just ‘impact’, instead of ‘impact to’.

c). Misuse of acronym. The acronym, MRI, is explained three times in the paper, whereas it should only be explained once at its first occurrence.

d). Misuse of verbs. The ‘localized’ used in line 80 and 89 should be replaced by ‘located’. ‘be located in/at/on/within’ is usually used to show the location of something.

e). Improper reference. As shown in lines 104-105 and line 342, the document should be cited as other references.

f). Typos. Line 236, ‘354.845 people….’

2 Paper structure

The overall structure of this paper is good, as it includes all the components of a research paper, including introduction, literature review, methodology, results, discussion, and conclusion. However, for some components, contents are not well organized. For example,

a). The discussion section is difficult to follow as it occupies two pages but without any subtitle.

b). The conclusion section should be improved to highlight the findings and contributions of this study.

3 Methodology

a). Papers reviewed by authors should be included as references. The authors of this paper have conducted a literature review, during which, as claimed in line 64, at least 28 papers were reviewed, but not all of them are cited.

b). The study area should be described in the methodology section. Use a map to show the study area and its location.

c). Revise the maps, i.e., figure 1 and figure 2, to make sure each map at least has a north arrow showing the north direction, a legend indicating what is included in the map, and a scale bar showing the scale of the map. And figures should be placed near where they are cited in the text.

d). The experiment and result should be better presented. In order to do this, numerous figures/maps are needed, including a map showing the location of waste sites being studied, figures showing the two layers described in line 191-192, and a map showing the census-tract.

e). The selection of the buffer, 100m, is subjective. The authors should discuss the possible influence of the selection of the buffer on the experiment result. A suggestion is to use different buffers to assess the influence of waste sites on the population.

4 Others

a). The meaning of line 212 is not clear.

b). Line 234, line 235, and other similar sentences should be removed or merged with other paragraphs, as it is not common for a short sentence to be a paragraph.

Author Response

  • Reviewer 1

Comments and Suggestions for Authors.

This paper developed a GIS-based approach for assessing the impact of waste sites on the nearby population. With this approach, the population and regions affected by waste sites were identified by using a municipal risk index. Overall, this paper is well presented. A solid background has been provided, a reasonable methodology has been used, and the results have been interpreted. However, there are still some problems regarding writing, paper structure, and methodology. The detailed comments are as follows:

1 Writing

a). Too many long and complex sentences, which affect the readability of this paper, such as lines 173-176 and lines 366-370

AU: THE STRUCTURE OF THE PAPER HAS BEEN REVISED AND SOME SENTENCES HAVE BEEN REWRITTEN. ENGLISH LANGUAGE REVIEW WAS OFFERED BY THE GUEST EDITOR IN INVITATION TO PRESENT THE PAPER, IN THE CASE IT WOULD BE ACCEPTED.

LINES 173-176: THE CONVERSION OF ALPHA-NUMERIC INDEX TO NUMERIC HAZARD RISK LEVEL HAS BEEN EXPLAINED IN NEW SUB-SECTION 2.1.4 (LINES 183-196)

LINES 366-370: CONCLUSION SECTION HAS BEEN REWRITTEN (LINES 467-488)

b). Minor issues with grammar and the use of collocations. For example, line 47-48, ‘illegal waste practices and …, impact to specific areas and populations [7].’, either use ‘impact on’ or just ‘impact’, instead of ‘impact to’.

AU: THE SENTENCES HAVE BEEN MODIFIED, FOLLOWING THE REVIEWERS’ COMMENT.

LINES 47-48: “to” PREPOSITION HAS BEEN REMOVED

c). Misuse of acronym. The acronym, MRI, is explained three times in the paper, whereas it should only be explained once at its first occurrence.

AU: THE EXPLANATION OF THE ACRONYM WAS LEFT IN LINE 271. SUBSEQUENTLY, ONLY THE ACRONYM HAS BEEN REPORTED.

d). Misuse of verbs. The ‘localized’ used in line 80 and 89 should be replaced by ‘located’. ‘be located in/at/on/within’ is usually used to show the location of something.

AU: THE TERM “LOCALIZED” HAS BEEN REPLACED BY “LOCATED” IN ALL PARTS OF THE TEXT.

e). Improper reference. As shown in lines 104-105 and line 342, the document should be cited as other references.

AU: THE DOCUMENT HAS BEEN INCLUDED IN REFERENCE’S LIST: REF. 22

f). Typos. Line 236, ‘354.845 people….

AU: THE NUMBER HAS BEEN CORRECTED: LINE 297

2 Paper structure

The overall structure of this paper is good, as it includes all the components of a research paper, including introduction, literature review, methodology, results, discussion, and conclusion. However, for some components, contents are not well organized. For example,

a). The discussion section is difficult to follow as it occupies two pages but without any subtitle.

AU: DISCUSSION SECTION HAS BEEN REVISED (LINES 337-466)

b). The conclusion section should be improved to highlight the findings and contributions of this study.

AU: CONCLUSION SECTION HAS BEEN REWRITTEN, HIGHLITHING THE FINDINGS AND CONTRIBUTIONS OF THE STUDY (LINES 467-488)

3 Methodology

a). Papers reviewed by authors should be included as references. The authors of this paper have conducted a literature review, during which, as claimed in line 64, at least 28 papers were reviewed, but not all of them are cited.

AU: REFERENCES OF SOME PAPERS INCLUDED IN THE REVIEW HAVE BEEN REPORTED AND ADDED IN THE REFERENCES’ LIST (REF 11-17). CONSIDERING THAT THE REVIEW (FAZZO ET AL, 2017) INCLUDED MORE THAN 60 PAPERS, SOME OF THESE HAVE BEEN MENTIONTED AS EXAMPLES (LINES 66-70)

b). The study area should be described in the methodology section. Use a map to show the study area and its location.

AU: THE MAP OF STUDY AREA HAS BEEN INCLUDED (FIGURE 1, PAGE 3)

c). Revise the maps, i.e., figure 1 and figure 2, to make sure each map at least has a north arrow showing the north direction, a legend indicating what is included in the map, and a scale bar showing the scale of the map. And figures should be placed near where they are cited in the text.

AU: NEW MAPS HAVE BEEN INCLUDED, REPORTING THE REQUIRED INFORMATION (NORTH ARROW, LEGEND, SCALE BAR): FIGURE 2-6. THE FIGURES HAVE BEEN PLACED IN THE TEXT AND SEPARATE JPG FILES HAVE BEEN UPLOADED.

d). The experiment and result should be better presented. In order to do this, numerous figures/maps are needed, including a map showing the location of waste sites being studied, figures showing the two layers described in line 191-192, and a map showing the census-tract.

AU: NEW MAPS HAVE BEEN INCLUDED: FIGURE 1: AREA AT STUDY (PAGE 3); FIGURE 2: WASTE SITES IN THE STUDY AREA (PAGE 7); FIGURE 3: THE STEPS FOR CREATING THE BREAK BUFFER OVERLAPS UP (DETAILED VIEW) (PAGE 8); FIGURE 4: CENSUS-TRACTS IN THE STUDY AREA (PAGE 8); FIGURE 5: OVERLAPS OF WASTE SITES IMPACT AREAS AFTER THE UNION BETWEEN THE CENSUS TRACTS LAYER AND THE BREAK BUFFER OVERLAPS UP LAYER. DETAILED VIEW IN THE PURPLE FRAME (PAGE 9); FIGURE 6: DISTRIBUTION OF MUNICIPAL RISK INDEX (MRI) CLASSES BY MUNICIPALITY (PAGE 14).

e). The selection of the buffer, 100m, is subjective. The authors should discuss the possible influence of the selection of the buffer on the experiment result. A suggestion is to use different buffers to assess the influence of waste sites on the population.

AU: THE EXPLANATION OF THE USE OF 100 METERS BUFFER HAS BEEN INCLUDED IN MATERIAL & METHODS SECTION (LINE 226-231) AND IN DISCUSSION THE INFLUENCE OF THIS CHOICE WAS DISCUSSED (LINE 350-353)

4 Others

a). The meaning of line 212 is not clear.

AU: FIGURE 2 HAS BEEN REPLACED AND THE OLD TEXT IN LINE 212 HAS BEEN REMOVED

b). Line 234, line 235, and other similar sentences should be removed or merged with other paragraphs, as it is not common for a short sentence to be a paragraph.

AU: THE STRUCTURE OF THE TEXT HAS BEEN REVISED. SHORT SENTENCES HAVE BEEN MERGED IN ONE PARAGRAPH.

THE SENTENCES IN 234-235 LINES HAVE BEEN MERGED IN ONE PARAGRAPH (SEE NEW 295-295 LINES).

Reviewer 2 Report

  • The topic of the paper is interesting and the paper is properly structured. However, since there are many paragraphs with a small number of sentences or a short content, the authors need to correct them.

  • I conclude that this manuscript is suitable for publication AFTER MINOR REVISION in International Journal of Environmental Research and Public Health. Reviewers' suggestions may be of assistance to you in the preparation of revised papers.

  1. [Abstract] A couple of words which show the section of the paper such as “Background”, “Methods”, “Results”, and “Conclusions” can be omitted in the abstract.

  1. [L103-105] I suggest authors to exclude the specific website in line 105 and (instead) fill a citation “[15]”. The website in line 105 can be listed in Reference section

  1. [Table 1] Is there any reason to describe the content (below the Waste Site in Table 1) in capital letters only? If there are no problems, please change the sentence to lower case to enhance the readability.

  1. [Section 2.1] I suggest the authors to insert several bullets to structure similar types of sentences such as the numbers and letters attributed to the corresponding modality of waste disposal and characteristics of the site / type of waste.

  1. [Table 2] What is the basis or evidence for the conversion of the HI value from Table 2 into numerical correspondence (values) such as 1, 10, 100, 1000, and 10000 in the relative potential waste impact category? In other words, it is necessary to explain what evidence that HI of 2 has 10 times the potential of HI of 1. This is because these absolute values have a great influence on the expression of the relative values in the final result map.

  1. [Table 2] In the same reason above, the authors need to explain the reason why A-G of Alpha-Part of HI can be "quantitatively" converted into a numeric correspondence in terms of relative potential waste impact.

  1. [L197-207] The 11 lines are composed of 5 paragraphs in this part. I recommend the authors to reduce the number of paragraphs considering the scope and relationship of the sentence content.

  1. [Figure 1] What is the meaning of the zone highlighted in sky-blue colored in Figure 1?

  1. [Figure 2 and Table 3] Why is the MRI value divided by 4 on the final results map? If these values are absolute values, there is no big problem in classification, but if they indicate relative risk, the number of class may need to be higher.

  1. [L341-342] I suggest authors to exclude the specific website in line 342 and (instead) fill a citation “[New number]”. The website in line 342 can be listed in Reference section

Author Response

  • Reviewer 2

Comments and Suggestions for Authors

  • The topic of the paper is interesting and the paper is properly structured. However, since there are many paragraphs with a small number of sentences or a short content, the authors need to correct them.
  • I conclude that this manuscript is suitable for publication AFTER MINOR REVISION in International Journal of Environmental Research and Public Health. Reviewers' suggestions may be of assistance to you in the preparation of revised papers.
  1. [Abstract] A couple of words which show the section of the paper such as “Background”, “Methods”, “Results”, and “Conclusions” can be omitted in the abstract.

AU: THE ABSTRACT HAS BEEN MODIFIED, FOLLOWING THE REVIEWER’S COMMENT (LINE 15-29)

  1. [L103-105] I suggest authors to exclude the specific website in line 105 and (instead) fill a citation “[15]”. The website in line 105 can be listed in Reference section

              AU: THE DOCUMENT HAS BEEN INCLUDED IN REFERENCES’ LIST: REF 22

  1. [Table 1] Is there any reason to describe the content (below the Waste Site in Table 1) in capital letters only? If there are no problems, please change the sentence to lower case to enhance the readability.

AU: THE SENTENCES IN TABLE 1 HAVE BEEN CHANGED TO LOWER CASE (TABLE 1, PAGE 5)

  1. [Section 2.1] I suggest the authors to insert several bullets to structure similar types of sentences such as the numbers and letters attributed to the corresponding modality of waste disposal and characteristics of the site / type of waste.

 AU: THE SECTION HAS BEEN DIVIDED IN SUB-SECTIONS: 2.1.1. (LINES 147-158); 2.1.2. (LINES 160- 174); 2.1.3.(LINES 176-181); 2.1.4 (LINES 183-196).

  1. [Table 2] What is the basis or evidence for the conversion of the HI value from Table 2 into numerical correspondence (values) such as 1, 10, 100, 1000, and 10000 in the relative potential waste impact category? In other words, it is necessary to explain what evidence that HI of 2 has 10 times the potential of HI of 1. This is because these absolute values have a great influence on the expression of the relative values in the final result map.

AU: DETAILS OF THE EVALUATION ON WHICH THE CONVERSION IS BASED HAVE BEEN INCLUDED IN 2.1.4 SUBSECTION OF MATERIALS AND METHODS SECTION (LINES 183-196)

  1. [Table 2] In the same reason above, the authors need to explain the reason why A-G of Alpha-Part of HI can be "quantitatively" converted into a numeric correspondence in terms of relative potential waste impact.

AU: DETAILS OF THE EVALUATION ON WHICH THE CONVERSION OF ALPHA-NUMERIC INDEX TO NUMERIC INDEX IS BASED HAVE BEEN INCLUDED IN 2.1.4 SUBSECTION OF MATERIALS AND METHODS SECTION (LINES 183-196):

2.1.4. Conversion of alpha-numeric Index to numeric Hazard risk level.

The next step was to derive a hazard quantification (Hazard Risk Level: HRL) from alpha-numeric index. The experts’ panel attributed a numeric Hazard risk level to each HI, considering that the magnitude of the impact of the sites depends on the combination between the type of waste (the letters of HI index) and the modalities of disposal and characteristics of the site (the number of HI). The attribution of the numeric correspondence was based on likelihood of the contamination by toxic agents and their potential hazard for the population. The most weight was attributed to the numeric part of HI, one order of magnitude moving from one to the next of the five levels, while letters were transformed in numbers without any amplification through different levels (Table 2). The conversion of the numeric part in ten powers was applied in such a way that the presence in one area of multiple low-hazard sites produced a hazard level lower than one high-hazard site (i.e. an illegal burning waste site is considered by far more impacting than many controlled landfills of inert materials). The value of the hazard risk level (HRL) is the product between the two numeric correspondences (i.e., an HI score equal to 5A corresponds to a hazard risk level of 60,000: 10,000*6).

  1. [L197-207] The 11 lines are composed of 5 paragraphs in this part. I recommend the authors to reduce the number of paragraphs considering the scope and relationship of the sentence content.

AU: THE STRUCTURE OF THE PAPER HAS BEEN REVISED AND THE NUMBER OF THE PARAGRAPHS HAS BEEN REDUCED.

LINES 197-207. THE TEXT HAS BEEN REVISED: SEE 2.2. SUBSECTION OF MATERIALS AND METHODS (LINES 209-289)

  1. [Figure 1] What is the meaning of the zone highlighted in sky-blue colored in Figure 1?

AU: FIGURE 1 HAS BEEN REMOVED

  1. [Figure 2 and Table 3] Why is the MRI value divided by 4 on the final results map? If these values are absolute values, there is no big problem in classification, but if they indicate relative risk, the number of class may need to be higher.

AU: THE CHOICE OF 4 CLASSES CATEGORIZATION HAS BEEN EXPLAINED IN METHODS SECTION (LINES 284-289) AND DISCUSSED IN DISCUSSION (LINES 420-432):

LINES 284-289. Finally, municipalities were categorized in four classes of risk (1 - low to 4 - high), on the basis of MRI, using Jenks method (Natural breaks), that maximizes homogeneity within groups and variance between groups. The categorization of municipalities in 4 MRI classes was evaluated the most appropriate, with respect to 5 or more classes, to distinguish the municipalities at the highest MRI, after a sensitivity analysis considering the distribution of municipalities and residing population by MRI classes.

LINES 420-432.The Hazard Index, and therefore the classes of the related MRI, is based on expert judgment with rare availability of measurements of pollutants to validate the choice of the categories. This process might be unable to quantify differences in terms of specific hazards. However, it allows us to distinguish among different likely hazardous exposure scenarios and to highlight municipalities differently impacted by waste, when looking at the health profile of populations residing in these areas.  HRL score classes represent a “relative” score, pointing out municipalities with the highest waste impact among those at study, and the municipalities included in the lowest HRL score class (class 1) are also impacted by waste sites (Table 4). The present categorization in 4 classes makes out, in particular, the municipalities at the highest waste impact (4 and 3 MRI classes), the principal aim of the present investigation. A different categorization of the absolute value of MRI is possible.

  1. [L341-342] I suggest authors to exclude the specific website in line 342 and (instead) fill a citation “[New number]”. The website in line 342 can be listed in Reference section

AU: THE MENTION OF THE WEBSITE HAS BEEN REMOVED AND THE REFERENCE HAS BEEN INCLUDED IN THE REFERENCES’ LIST (REF 22)

Reviewer 3 Report

Overall a very good and very worthwhile research paper. It is also quite impressive how relevant and strong the team behind this study: Ministry of health and of the environment, including a court magistrate, who requested the study in the first place.

A few “less minor” issues like:

  • Figure 1 – does not make much sence
  • I must admit that I initially found the “ranking system” a bit counter-intuitve. by both letter and number that the HI i
  • This being a GIS I would have expected a bit more visual representation in the form of Figures, and maps.
  • At the linquistic level, the paper is also rather well structured and easy to understand. However there are quite a few “minor” (but cumulative) grammatical edits that are necessary to make it an even better paper. Thus, I recommend that the authors consult with a native (or near native) English language speaker, to improve those few run-on senteces, or preposition “mis-position”. I would gladly  provide them with a copy of a more detailed review, or be happy to read it one more time before publishing.

These are some of the recurrent “syntax and grammar” issues that should be improved. Prepositions and “run-on” sentences that become paragraphs are common hurdles for most non-native speakers, of any language.

  1. Small inconsistencies such as “metre” abbreviated differently at different parts in the text (i.e.: m versus mt)
  1. Misuse of prepositions or adverbs in some instances, and general writing structure:
    • in the abstract the word “notwithstanding” is erroneously used
    • colloquial use of the therm “anyways” on line 328
    • general (minor but critical) preposition mixups
    • Word order in sentences or sentence structure across paragraph, like in line 240, 254 and 255 for the former and line 251 for the latter
    • lack of the article when not needed in italian, but needed in english (and vice versa)
    • Writing structure: paragraph starting line 321 and Line 328 seems to «colloqual_ anywais»!
    • Very long sentences that make up an actual paragraph, like the one starting 326: this sentence can/should be broken up in more than one
  2. A few paragraphs/sentences need immediate attention, as they either “don’t make sense” or they’re so complex that the word order only adds more to it:
    • th use of the word “nothwithstanding” in the abstract.
    • The first paragraph in the Results is a good example of grammatical imrpovements such as replacing, and others like this:
    • Line 269 and 289: allow to identify, rather should be either «allows us to identify or allows the identificationanyhow (lines 293-294)

Author Response

  • Reviewer 3

Comments and Suggestions for Authors

Overall a very good and very worthwhile research paper. It is also quite impressive how relevant and strong the team behind this study: Ministry of health and of the environment, including a court magistrate, who requested the study in the first place.

A few “less minor” issues like:

  • Figure 1 – does not make much sence

AU: FIGURE 1 HAS BEEN REMOVED

  • I must admit that I initially found the “ranking system” a bit counter-intuitve. by both letter and number that the HI i

AU: THE METHODS USED TO THE ATTRIBUTION OF HAZARD RISK LEVEL INDEX BY WASTE SITE HAVE BEEN SPECIFIED (LINES 132- 196)

  • This being a GIS I would have expected a bit more visual representation in the form of Figures, and maps.

AU: NEW MAPS HAVE BEEN INCLUDED: FIGURE 1: AREA AT STUDY (PAGE 3); FIGURE 2: WASTE SITES IN THE STUDY AREA (PAGE 7); FIGURE 3: THE STEPS FOR CREATING THE BREAK BUFFER OVERLAPS UP (DETAILED VIEW) (PAGE 8); FIGURE 4: CENSUS-TRACTS IN THE STUDY AREA (PAGE 8); FIGURE 5: OVERLAPS OF WASTE SITES IMPACT AREAS AFTER THE UNION BETWEEN THE CENSUS TRACTS LAYER AND THE BREAK BUFFER OVERLAPS UP LAYER. DETAILED VIEW IN THE PURPLE FRAME (PAGE 9); FIGURE 6: DISTRIBUTION OF MUNICIPAL RISK INDEX (MRI) CLASSES BY MUNICIPALITY (PAGE 14).

  • At the linquistic level, the paper is also rather well structured and easy to understand. However there are quite a few “minor” (but cumulative) grammatical edits that are necessary to make it an even better paper. Thus, I recommend that the authors consult with a native (or near native) English language speaker, to improve those few run-on senteces, or preposition “mis-position”. I would gladly  provide them with a copy of a more detailed review, or be happy to read it one more time before publishing.
  1. THE TEXT HAS BEEN REVISED. WE WOULD PLEASED TO BENEFIT OF THE FREE ENGLISH EDITING, IN THE CASE THE MANUSCRIPT IS ACCEPTED, THAT THE GUEST EDITOR (Dr BERMANN) OFFERED IN THE INVITATION TO PARTICIPATE TO THE SPECIAL ISSUE.

These are some of the recurrent “syntax and grammar” issues that should be improved. Prepositions and “run-on” sentences that become paragraphs are common hurdles for most non-native speakers, of any language.

  1. Small inconsistencies such as “metre” abbreviated differently at different parts in the text (i.e.: m versus mt)

AU: “METRE” TERM HAS BEEN ABBREVIATED IN m, IN ALL PARTS OF THE TEXT

  1. Misuse of prepositions or adverbs in some instances, and general writing structure:
  • in the abstract the word “notwithstanding” is erroneously used.

AU: THE TERM HAS BEEN REMOVED (LINE 16)

  • colloquial use of the therm “anyways” on line 328.

AU: THE TERM HAS BEEN REMOVED AND THE SENTENCE HAS BEEN REWRITTEN (LINES: 426-432)

  • general (minor but critical) preposition mixups

AU: SOME PREPOSITIONS HAVE BEEN REMOVED

  • Word order in sentences or sentence structure across paragraph, like in line 240, 254 and 255 for the former and line 251 for the latter

AU: THE STRUCTURE OF THE PAPER HAS BEEN REVISED AND SOME SENTENCES HAVE BEEN REWRITTEN. ENGLISH LANGUAGE REVIEW WAS OFFERED BY THE GUEST EDITOR IN THE INVITATION TO PRESENT THE PAPER, IN THE CASE THAT IT WOULD BE ACCEPTED.

  • lack of the article when not needed in italian, but needed in english (and vice versa)

AU: THE GRAMMAR OF THE TEXT HAS BEEN REVISED

  • Writing structure: paragraph starting line 321 and Line 328 seems to «colloqual_ anywais»!

AU: THE TERM HAS BEEN REMOVED AND THE SENTENCE HAS BEEN REWRITTEN (LINES 413-432)

  • Very long sentences that make up an actual paragraph, like the one starting 326: this sentence can/should be broken up in more than one

AU: THE SENTENCES HAVE BEEN REWRITTEN AND THE SENTENCE STARNTIN 326 LINE HAS BEEN BROKEN UP IN MORE SENTENCES (LINES 423-430)

  1. A few paragraphs/sentences need immediate attention, as they either “don’t make sense” or they’re so complex that the word order only adds more to it:

AU: THE STRUCTURE AND THE GRAMMAR OF THE PAPER HAVE BEEN REVISED AND SOME SENTENCES HAVE BEEN REWRITTEN. ENGLISH LANGUAGE REVIEW WAS OFFERED BY THE GUEST EDITOR IN THE INVITATION TO PRESENT THE PAPER, IN THE CASE THAT IT WOULD BE ACCEPTED.

  • the use of the word “nothwithstanding” in the abstract.

AU: IN THE NEW ABSTRACT THE WORD HAS BEEN REMOVED (LINE 16)

  • The first paragraph in the Results is a good example of grammatical imrpovements such as replacing, and others like this:
  • Line 269 and 289: allow to identify, rather should be either «allows us to identify or allows the identificationanyhow (lines 293-294)

AU: THE SENTENCES HAVE BEEN REWRITTEN (LINES: 297, 338, 423).

Round 2

Reviewer 1 Report

Thanks for the revised manuscript. Many of the problems in the previous version have been solved. For example, figures have been added to better illustrate the study. However, not all of these problems have been properly solved. The detailed comments are as follows.

1 Writing and language

a) What is HI short for, Hazard Index (line170) or Hazardous Index (line 220)? Please be consistent.

b) line 179: grammar error. “…, the corresponding HI raised up”

c) line 135-136: “The sites present in … have been reported.” The meaning is not clear.

d) Figure 1, Figure 2, Figure 4, and others. Remove the “Figure X” from the map title, and an independent caption should be added below each figure.

e) Remove the comma, “,”, from numbers, use a single space instead if needed.

Professional proofreading is suggested to improve the quality of this paper.

2 Paper structure

a) The study area is suggested to be introduced in section 2 as an independent subsection. Move all the figures and descriptions of the study area, such as line 357, to that subsection.

b) The discussion section should be more organized. There are mainly five parts contained in the current discussion section, including (a) the main findings of this study, (b) discussion on the selection of buffer, (c) discussion on the limitations of this study, (d) discussion on MRI, and (e) contribution of this study. The suggestion is to move (a) to the conclusion section and assign a subtitle to each of the remaining parts and discuss them in detail.

c) Conclusion section. Use 1-2 sentences to briefly introduce the study, including the problem faced and the method used; then described the main findings and limitations of this study, and finally point out the potential direction of future work.

3 Methodology

a). The previous comment on the selection of the buffer has not been properly addressed, even though the reason for using 100 meters as buffer distance has been explained. In the previous round of review, it was suggested to use different buffers to assess the influence of waste sites on the population, as it involves a well-recognized issue in the geospatial area, i.e. scale issue, which means the use of different scales (or buffers in the case of this study) will lead to different results. If this comment cannot be addressed in this study, at least discuss it in the discussion section.

Author Response

REVIEWER 1- SECOND ROUND

Thanks for the revised manuscript. Many of the problems in the previous version have been solved. For example, figures have been added to better illustrate the study. However, not all of these problems have been properly solved. The detailed comments are as follows.

1 Writing and language

  1. What is HI short for, Hazard Index (line170) or Hazardous Index (line 220)? Please be consistent.

AU: THE “HAZARDOUS” INDEX TERM HAS BEEN CHANGED INTO “HAZARD” INDEX: LINE 216, LINE 220 (TITLE OF TABLE 1)

  1. line 179: grammar error. “…, the corresponding HI raised up”

AU: THE VERBAL FORM HAS BEEN CORRECTED: “THE CORRESPONDING HI HAS BEEN RAISED UP” (LINE 184)

  1. line 135-136: “The sites present in … have been reported.” The meaning is not clear.

AU: THE VERB HAS BEEN CHANGED: “THE SITES IN .. HAVE BEEN TRANSCRIBED” (LINE 151)

  1. Figure 1, Figure 2, Figure 4, and others. Remove the “Figure X” from the map title, and an independent caption should be added below each figure.

AU: THE TITLE HAS BEEN REMOVED FROM ALL FIGURE FILES AND AN INDEPENDENT WORD FILE REPORTING THE TEXT OF THE TITLE OF EACH FIGURE HAS BEEN SENT. IN THE MANUSCRIPT THE TITLE HAS BEEN WRITTEN BELOW EACH FIGURE (lines 119, 124, 153, 256, 278, 351)

  1. Remove the comma, “,”, from numbers, use a single space instead if needed.

AU: THE COMMA HAVE BEEN REMOVED FROM THE NUMBERS.

Professional proofreading is suggested to improve the quality of this paper.

2 Paper structure

  1. The study area is suggested to be introduced in section 2 as an independent subsection. Move all the figures and descriptions of the study area, such as line 357, to that subsection.

AU: THE SUBSECTION 2.1 “THE STUDY AREA” HAS BEEN INCLUDED IN SECTION 2 (LINES 92-167). FIGURE 1 (THE STUDY AREA), THE FORMER FIGURE 4 (MUNICIPALITIES AND CENSUS-TRACTS AT STUDY), AND THE MAP OF THE WASTE SITES (FIGURE 3) HAVE BEEN REMOVED TO THIS SUBSECTION. THE OTHER FIGURES HAVE BEEN LEFT IN THE SECTION WHERE THEY ARE MENTIONED.

  1. The discussion section should be more organized. There are mainly five parts contained in the current discussion section, including (a) the main findings of this study, (b) discussion on the selection of buffer, (c) discussion on the limitations of this study, (d) discussion on MRI, and (e) contribution of this study. The suggestion is to move (a) to the conclusion section and assign a subtitle to each of the remaining parts and discuss them in detail.

AU: THE DISCUSSION SECTION (LINES 359-455) HAS BEEN REORGANIZED AND SUBTITLE HAS BEEN ASSIGNED TO THE DIFFERENT PARTS. THE MAIN RESULTS HAVE BEEN LEFT ALSO IN THIS SECTION, BECAUSE, IN OUR OPINION, THIS PART CONTRIBUTES TO THE DISCUSSION OF THE RESULTS.

  1. Conclusion section. Use 1-2 sentences to briefly introduce the study, including the problem faced and the method used; then described the main findings and limitations of this study, and finally point out the potential direction of future work.

AU: AN INTRODUCTORY PARAGRAPH  (LINES 457-461) AND A SENTENCE REGARDING THE FURTHER EPIDEMIOLOGICAL STUDY IN THE AREA HAVE BEEN INCLUDED (LINES 472-473). FOLLOWING THE AUTHORS’ GUIDELINES OF THE REVIEW, THE CONCLUSION SECTION WOULD BE CONCISE AND, IN OUR OPINION, TO ADD OTHER SENTENCES WOULD NOT COMPLY WITH THE AUTHORS’ GUIDELINES. BELOW THE NEW CONCLUSIONS SECTION (LINES 456-477).

  1. Conclusions

The present investigation was inspired by and originated from a formal request to North Naples Prosecutor Office to the National Institute for Health, to identify the municipalities at the highest waste impact in an area characterized by a widespread waste mismanagement and illegal practices. The ad hoc developed GIS-approach, notwithstanding the limitation described in the paper, identified the municipalities affected by different waste impact levels.

In the whole area there is a high environmental pressure by waste sites and likely human exposure to a variety of agents, including toxic ones.  Thirty-eight percent of population lives within 100 meters from one or more waste sites. Illegal and uncontrolled waste sites, including 653 waste burning sites, represent about 90% of all 2767 waste sites present in the area. Municipalities with different levels of predicted waste exposure risk for the population were identified with the GIS approach.

On the basis of the present results, environmental remediation actions and stopping still ongoing illegal and poor waste management practices are urgently needed. The data reported in the GIS system provide useful information to the Prosecutor, in order to contrast illegal waste mismanagements and to prosecute the criminal acts regarding waste trafficking and management.

The municipal GIS indicator will be applied in further epidemiological study on the correlation between waste impact and the occurrence of specific diseases, in population residing in study area.

The GIS method described in the present paper could be usefully applied also in the lack of detailed environmental data and analytical information on waste sites. These situations could be common in similar contexts, of informal and illegal waste management, in specific areas of both industrialized and in middle-low income countries. 

3 Methodology

a). The previous comment on the selection of the buffer has not been properly addressed, even though the reason for using 100 meters as buffer distance has been explained. In the previous round of review, it was suggested to use different buffers to assess the influence of waste sites on the population, as it involves a well-recognized issue in the geospatial area, i.e. scale issue, which means the use of different scales (or buffers in the case of this study) will lead to different results. If this comment cannot be addressed in this study, at least discuss it in the discussion section.

AU: THE CHOICE OF THE USE OF 100 METERS BUFFER HAS BEEN BETTER EXPLAINED IN METHODS (LINES 259-265) AND IN DISCUSSION SECTIONS: IN THE LATTER, A SUBSECTION “SELECTION OF WASTE IMPACT AREA BUFFER” HAS BEEN ADDED (LINES 371-381).  THE CHOICE WAS ESSENTIALLY BASED ON A PRIORI CONSIDERATIONS. ONE ALTERNATIVE APPROACH MIGHT HAVE BEEN THAT OF SELECTING THE RADIUS AFTER TESTING THE EFFECTS OF THE ADOPTION OF DIFFERENT VALUES FOR THIS PARAMETER. IN OUR OPINION, GIVEN THE CONTEXT AND THE AIMS OF THIS STUDY, NAMELY, TO PROVIDE ELEMENTS FOR PRIORITY EVALUATION IN ADOPTING SITE-SPECIFIC REMEDIAL ACTION, IT WAS AGREED TO FOLLOW THE A PRIORI RATHER THAN A POSTERIORI APPROACH. MOREOVER, THE APPLICATION OF A LOW RADIUS WASTE IMPACT AREAS ALLOWED US TO DISTINGUISH THE SUBAREAS AT DIFFERENT LEVELS OF WASTE IMPACT AND TO HIGHLIGHT THE HIGHEST IMPACTED AREAS, PREVENTING FALSE POSITIVES. BELOW THE NEW SUBSECTIONS.

METHODS SECTION (LINES 220-222). Buffers of larger breadth, had caused an overlap of the impacted areas and whole area would appear to be interested by waste impact, preventing a distinction of the areas at different degrees of waste impact.

DISCUSSION SECTION (LINES 371-384): The rationale for choosing a 100 m radius buffer was illustrated in the Methodology section. The choice was essentially based on a priori considerations. One alternative approach might have been that of selecting the radius after testing the effects of the adoption of different values for this parameter. Given the context and the aims of this study, namely, to provide elements for priority evaluation in adopting site-specific remedial action, it was finally agreed to follow the a priori rather than a posteriori approach. Moreover, the application of a low radius waste impact areas allows us to distinguish the subareas at different levels of waste impact and to highlight the highest impacted areas, preventing false positives.
